# Progression of Thoracic Aortic Dissection Is Aggravated by the hsa_circ_0007386/miR-1271-5P/IGF1R/AKT Axis via Induction of Arterial Smooth Muscle Cell Apoptosis

**DOI:** 10.3390/biomedicines11020571

**Published:** 2023-02-15

**Authors:** Xinsheng Xie, Xiang Hong, Shichai Hong, Yulong Huang, Gang Chen, Yihui Chen, Yue Lin, Weifeng Lu, Weiguo Fu, Lixin Wang

**Affiliations:** 1Department of Vascular Surgery, Xiamen Branch, Zhongshan Hospital, Fudan University, Xiamen 361015, China; 2Department of Vascular Surgery, Vascular Surgery Institute of Fudan University, Zhongshan Hospital, Fudan University, Shanghai 200032, China

**Keywords:** thoracic aortic dissection, circRNA, miRNA, mRNA, IGF1R

## Abstract

Background: The molecular mechanisms associated with thoracic aortic dissection (TAD) remain poorly understood. A comprehensive high-throughput sequencing-based analysis of the circRNA–miRNA–mRNA competitive endogenous RNA (ceRNA) regulatory network in TAD has not been conducted. The purpose of this study is to identify and verify the key ceRNA networks which may have crucial biological functions in the pathogenesis of TAD. Methods: Gene expression profiles of the GSE97745, GSE98770, and GSE52093 datasets were acquired from the Gene Expression Omnibus (GEO) database. Differentially expressed genes (DEGs) were identified using the GEO2R tools. Protein–protein interaction (PPI) networks of the hub genes were constructed using STRING; the hub genes and modules were identified by MCODE and CytoHubba plugins of the Cytoscape. We analyzed the hub genes using Gene Ontology and Kyoto Encyclopedia of Genes and Genomes pathway enrichment analysis. The functions of these hub genes were assessed using Cytoscape software. Our data—along with data from GSE97745, GSE98770, and GSE52093—were used to verify the findings. Results: Upon combined biological prediction, a total of 11 ce-circRNAs, 11 ce-miRNAs, and 26 ce-mRNAs were screened to construct a circRNA–miRNA–mRNA ceRNA network. PPI network and module analysis identified four hub nodes, including IGF1R, JAK2, CSF1, and GAB1. Genes associated with the Ras and PI3K-Akt signaling pathways were clustered in the four hub node modules in TAD. The node degrees were most significant for IGF1R, which were also the most significant in the two modules (up module and hub module). IGF1R was selected as a key gene, and the hsa_circ_0007386/miR-1271–5P/IGF1R/AKT regulatory axis was established. The relative expression levels of the regulatory axis members were confirmed by RT-PCR in 12 samples, including TAD tissues and normal tissues. Downregulation of IGF1R expression in smooth muscle cells (SMCs) was found to induce apoptosis by regulating the AKT levels. In addition, IGF1R showed high diagnostic efficacy in both AD tissue and blood samples. Conclusions: The hsa_circ_0007386/miR-1271-5P/IGF1R/AKT axis may aggravate the progression of TAD by inducing VSMCs apoptosis. CeRNA networks could provide new insights into the underlying molecular mechanisms of TAD. In addition, IGF1R showed high diagnostic efficacy in both tissue and plasma samples in TAD, which can be considered as a diagnostic marker for TAD.

## 1. Introduction

Aortic dissection (AD) refers to the tearing of the aorta at the intima from various causes, resulting in blood entering the middle layer of the aortic wall and gradually separating the middle layer to form both true and false cavities driven by blood pressure [1,2]. If AD occurs within type A dissections, 40% of patients die immediately and mortality is 1–2% for each hour afterwards, resulting in a 48-h mortality of approximately 50% [3]. In type B dissections, approximately 36–50% of patients with thoracic aortic dissection (TAD) will die of aortic rupture within 48 h, and 65–75% will die within two weeks; the one-year survival rate is less than 10% [4,5]. Presently, TAD diagnosis and treatment remain a challenge for vascular surgeons. The complex pathogenesis of TAD has not been fully elucidated. Therefore, clarifying the pathogenesis of TAD will help prevent and treat the occurrence and development of TAD, thereby reducing the incidence of the disease and improving the patient survival rate.

Studies have shown that more than 97% of the genome encodes non-coding transcripts, and most of these transcripts are processed into non-coding RNA, such as miRNA, lncRNA, circRNA, siRNA, and piRNA [6]. Non-coding RNAs (ncRNA) are RNAs that do not have protein-coding ability. In-depth research and understanding of the functions of ncRNAs suggest that they play a vital role in regulating gene expression, involving a variety of cellular physiological processes and leading to disease progression. Most studies suggest that non-coding RNAs mainly interact with DNA, mRNA, and proteins to perform biological functions [7]. Circular RNAs (circRNAs) are single-stranded closed-loop RNA molecules without terminal 5′ caps and 3′ poly (A) tails. Unlike other linear ncRNAs, circRNA forms a closed loop through a covalent bond between the 3′ and 5′ ends. It is widely present in humans and mice and is expressed with good stability, tissue specificity, and conservation [8,9,10]. The expression of circRNAs is regulated in tissues, and the developmental stage-specific expression suggests its functional relevance. Recently, studies have shown that circRNA can be used as a splicing and transcription regulator of parental genes [11,12]. In addition, some studies have proposed the mechanism of action of ceRNA: circRNA can competitively bind to the same miRNA, thus affecting the stability of mRNA or translation function [13]. Many circRNAs can be detected in a relatively stable manner in body fluids, and they display remarkable potential as disease biomarkers. Increasing evidence has shown that circRNAs may play a key role in disease progression [14,15]. miRNA is one type of endogenous ncRNA, with a length of approximately 22 bases. Studies have shown that on binding to mRNA, miRNA degrades or inhibits the translation of mRNA and participates in post-transcriptional regulation in animals and plants [16,17]. In the genome, about one-third of genes are regulated by miRNAs. In a wide range of biological activities and disease processes, miRNAs play an important role in the regulation of gene expression [18].

In this study, we used high-throughput sequencing databases to screen differentially expressed circRNAs, miRNAs, and mRNAs in TAD diseases. We also used the bioinformatics analysis method to construct a circRNA-mediated ceRNA regulatory network in TAD diseases and screens that may be highly related to TAD diseases. The circRNA-mediated ceRNA regulation axis was further verified to provide a new theoretical basis for the in-depth study of the molecular mechanism underlying TAD occurrence and development.

## 2. Materials and Methods

### 2.1. Reagents and Equipment

Antibodies against IGF1R (ab263903, Abcam, USA); AKT (ab8805, Abcam, Boston, MA, USA); CASPASE3 (ab4051, Abcam, USA) were purchased from Abcam (https://www.abcam.cn/; accessed on 16 September 2022). Anti-GAPDH ((14C10) Rabbit mAb #2118, USA) antibodies were from Cell Signaling Technology. Secondary antibodies (goat anti-rabbit Alexa 488 and 594) used for immunocytochemistry were from Life Technologies. ELISA kit (ab100546, Abcam, USA) and TUNEL Assay Kits (ab661100, Abcam, USA) were from Abcam. Total RNA was extracted from tissue samples using TRIzol^®^ reagent (Invitrogen; Thermo Fisher Scientific, Inc., Waltham, MA, USA).Confocal laser scanning microscope (Leica TSC SP8, Lycra GMBH, Wetzlar, Germany). GStorm Gradient PCR (MX3000P, Agilent Inc., Santa Clara, CA, USA). Molecular biological imager (ImageQuant LAS 4000 Mini, GE Corporation, Boston, MA, USA). SPSS software, version 23.0 (IBM Corp., Armonk, NY, USA), R software (https://www.r-project.org/), version 4.2.1.

### 2.2. Datasets and Analysis of Differentially Expressed Genes

The expression profiling datasets GSE97745 (circRNA, GPL21825), GSE98770 (microRNA, GPL17660), and GSE52093 (mRNA, GPL10558) were downloaded from the GEO database (https://www.ncbi.nlm.nih.gov/geo/; accessed on 12 August 2022) of NCBI. The GSE97745 dataset contained three TAD samples and three normal aortic samples, GSE98770 contained six TAD samples and five normal aortic samples, and GSE52093 contained seven TAD samples and five normal aortic samples. All samples were taken from the ascending aorta wall, avoiding the influence of different sampling sites on data analysis results. The experimental samples in our center were taken from aortic walls and blood samples of TAD patients and organ donors. The analysis was performed on the TAD tissue and serum at the time of emergency surgery for the dissection. Aortic specimens were collected from type A TAD patients undergoing aortic replacement in the intimal tear position at the hospital. Normal thoracic aortas (NA) were obtained from organ donors without aortic diseases. We identified DEGs using the GEO2R tools. Values with *p* < 0.05 and |log2Fold change| (|logFC|) > 1.0 were considered statistically significant.

### 2.3. Construction of the circRNA–microRNA–mRNA Network

CSCD (http://gb.whu.edu.cn/CSCD/; accessed on 15 August 2022) and circBase (http://www.circbase.org/; accessed on 15 August 2022) were used to predict miRNA-binding sites (MREs), which were considered as potential target miRNAs of the DEG circRNAs. In addition, three online databases—miRDB (http://mirdb.org/; accessed on 15 August 2022), Target Scan Human (http://www.targetscan.org/vert_72/; accessed on 15 August 2022), and miRTarbase (http://mirtarbase.mbc.nctu.edu.tw/index.html; accessed on 15 August 2022)—were used to predict mRNAs that were considered potential target mRNAs of the DEG miRNAs. Co-RNAs (co-miRNAs and co-mRNAs) were identified by Venn analysis between the DEG RNAs (DEG miRNAs and DEG mRNAs) and target RNAs (target miRNAs and target mRNAs), respectively. We used a Sankey diagram to illustrate the interactions of ce-circRNAs (ceRNA-circRNAs), ce-miRNAs (ceRNA-miRNAs) and ce-mRNAs (ceRNA-mRNAs). The interactive networks of ce-circRNAs, ce-miRNAs and ce-mRNAs were established and visualized using Cytoscape 3.7.2. The diagram of ceRNA construction is shown in Figure 1.

### 2.4. Generation of the PPI Network and the Functional Enrichment Analysis of Target Genes

The STRING database (http://string-db.org) was used to search for the PPI (Protein–Protein Interaction) pairs of co-mRNAs and ce-mRNAs in the TAD and normal aortic samples. A PPI network map was constructed using Cytoscape software. Meanwhile, molecular complex detection (MCODE) identified the most important module of the network map. PPI networks were generated using Cytoscape software (http://cytoscape.org/). A node in the PPI network denotes a protein, and edges denote interactions. The criteria for analysis were a degree cutoff of 2, MCODE scores > 5, max depth of 100, k-score of 2, and node score cutoff of 0.2. Hub genes were excavated by using OmicShare. We explored the biofunctions of the target genes by Gene Ontology (GO) enrichment and Kyoto Encyclopedia of Genes and Genomes (KEGG) pathways using the “Bioconductor” and “Cluster Profiler” package in R 3.6.1 software. Functional pathway analysis mapped genes to the KEGG pathways. A Benjamini-adjusted *p*-value of 0.05 was set as the cutoff for screening significant GO terms and the KEGG pathways.

### 2.5. Immunohistochemistry (IHC), TUNEL Staining and Western Blot Assay

Immunohistochemistry (IHC) assays and evaluation of immunohistochemistry intensity were performed as previously described [19]. TUNEL staining assay was performed according to the method steps provided by the TUNEL Assay Kits (ab66108, Abcam, USA). Western blot assays were performed as previously described [20]. Detailed methods are available in the Appendix A. Ethical approval was obtained from the Ethics Committee of the Xiamen Branch, Zhongshan Hhospital, Fudan University with the approval Code: B2021-718.

### 2.6. Statistical Analysis

Results are presented as mean ± standard error of the mean. Statistical significance of the differences between two groups was determined using the Student’s *t*-test. The expression of TAD hub genes was compared for correlation analysis using Pearson’s rho (ρ) tests. All data were statistically analyzed using SPSS software, version 23.0 (IBM Corp., Armonk, NY, USA). Values with *p* < 0.05 were considered significantly different.

## 3. Results

### 3.1. Identification of DEGs and Target Genes

Three microarray databases (GSE97745, GSE98770, and GSE52093) were included in our study. We analyzed the raw data using the GEO2R tool based on the following thresholds: *p* < 0.05 and |LogFC| > 1.0. From the GSE97745 dataset, we identified 66 DEG circRNAs. From the GSE98770 dataset, we identified 81 DEG miRNAs. From the GSE52093 dataset, we identified 1524 DEG mRNAs. The volcano plots show the DEG circRNAs (Figure 2A), DEG miRNAs (Figure 2B), and DEG mRNAs (Figure 2C) in the TAD and normal aortic samples. There were 30 co-miRNAs and 79 co-mRNAs identified by Venn analysis between the DEG miRNAs and target miRNAs (Figure 2D) and the DEG mRNAs and target mRNAs (Figure 2E).

### 3.2. Construction of the ceRNA Network

A total of 35 DEG circRNAs were selected for further study, for which a total of 1386 target miRNAs was predicted by CSCD and circBase. The target mRNAs for each of these 30 co-miRNAs were predicted using three online databases: miRDB, Target Scan Human, and miRTarbase. A total of 1323 mRNAs that appeared in all three databases were selected as target mRNAs. Finally, we chose 35 DEG circRNAs, 30 co-miRNAs, and 79 co-mRNAs for the construction of the ceRNA network. Heatmaps of these DEGs (11 ce-circRNAs, 11 ce-miRNAs, and 26 ce-mRNAs) from the ceRNA network used in hierarchy cluster analysis are shown (Figure 2F–H). The circRNA–miRNA–mRNA network is shown in Figure 3A,B.

### 3.3. Generation of the PPI Network and Enrichment Analysis of the Key Module

We established a PPI network of the 79 co-mRNAs and 26 ce-mRNAs using STRING and visualized it using Cytoscape (Figure 4A). We also analyzed modules using the Cytoscape plugin and MCODE and then identified the top modules (Figure 4B). The lists of the 79 co-mRNAs and the 26 ce-mRNAs both showed the same top modules (Figure 4C). The significant nodes of the top modules based on PPI network analysis were four hub nodes including IGF1R, JAK2, CSF1, and GAB1. The node degrees were the most significant for IGF1R, which were also the most significant in the two modules (respectively from module of 79 co-mRNAs and 26 ce-mRNAs, Figure 4D).

We analyzed the GO enrichment and KEGG pathways using the Cluster Profiler package in R to determine the function of the four nodes. Enrichment analyses showed that the top three terms in the biological process (BP) category included positive regulation of cell migration, platelet-derived growth factor receptor signaling pathway, and peptidyl-tyrosine autophosphorylation. The enriched terms in the cellular component (CC) category were related to the membrane protein complex. The enriched terms in the molecular function (MF) category in TAD mainly included insulin receptor substrate binding, phosphatidylinositol 3-kinase binding, and protein–tyrosine kinase activity (Figure 5A,B and Table 1). The KEGG pathway analyses of the four hub genes revealed that mRNAs were mainly involved in the Ras signaling pathway and PI3K–Akt signaling pathway (Figure 5C,D and Table 2).

### 3.4. The hsa_circ_0007386/miR-1271-5P/IGF1R/AKT Axis May Aggravate the Progression of TAD by Inducing VSMCs Apoptosis

Based on the four hub nodes and the enrichment analysis results, we constructed a mulberry map of ceRNAs associated with the pathways (Figure 6A). The expression levels of hsa_circ_0007386 and IGF1R were downregulated, whereas that of miR-1271-5P was upregulated in TAD (Figure 6B–D). The expression levels of the DEGs from the ceRNA network between the AD group and the control group are shown in Appendix A. Aortic wall tissues from eight TAD patients and five organ transplant donors were collected for RT-PCR to detect the mRNA levels of hsa_circ_0007386, miR-1271-5P, and IGF1R. Similarly, the expression levels of hsa_circ_0007386 and IGF1R were downregulated, whereas that of miR-1271-5P was upregulated in our TAD data (Figure 6E–G). In addition, we examined the protein levels of IGF1R in paraffin-embedded arterial wall tissue samples from five TAD patients. We found that the expression of IGF1R was downregulated in arterial wall tissues with dissection compared to that in arterial parietal tissue without dissection (Figure 7A). The score for IGF1R was higher in arterial parietal tissue without dissection (n = 5; 71.43%) than in the corresponding arterial wall tissues with dissection (Figure 7B). Subsequently, we used picropodophyllin (PPP), an IGF1R inhibitor, in the normal aortic smooth muscle cells (NASMCs) and found that PPP can induce apoptosis in NASMCs (Figure 7C). Both STRING [21] and GENEMANIA [22] databases showed that IGF1R interacts with AKT (Figure 7D and Appendix A). Western blot analysis showed that the expression levels of AKT and Caspase3 were increased in NASMCs, which downregulated IGF1R expression by PPP (Figure 7E).

### 3.5. IGF1R Can Be Considered as a Diagnostic Marker for TAD

The area under the ROC curve (*p* < 0.02, AUC = 0.91) indicates that the IGF1R have potential diagnostic value and might serve as biomarkers of TAD in GSE52093 (Figure 7F). To further verify the diagnostic efficacy of IGF1R, the IGF1R levels in plasma from 48 TADs and 40 healthy controls were measured by ELISA (Figure 7G and Appendix A). Our results show that the IGF1R levels in plasma were significantly increased in TAD patients compared with healthy controls. Similarly, IGF1R showed high diagnostic efficacy in AD blood samples (*p* < 0.01, AUC = 0.81, Figure 7H).

## 4. Discussion

In this study, high-throughput sequencing data were extracted from ascending aorta tissues of patients with TAD and normal controls to screen out differentially expressed circRNAs, miRNAs, and mRNAs. We created the network map of molecular and cellular interactions. Based on the four hub nodes and the enrichment analysis results, the hsa_circ_0007386/miR-1271–5P/IGF1R/AKT signal axis was finally selected for further verification research. We also analyzed the gene expression in samples of full aortic walls from patients without TAD (control) and with TAD to explore the potential clinical relationships between hub DEGs. The expression of hsa_circ_0007386 and IGF1R was downregulated, whereas that of miR-1271-5P was upregulated in TAD tissue. Similarly, the expression of hsa_circ_0007386 and IGF1R was downregulated, whereas that of miR-1271-5P was upregulated in our TAD tissue data. Different from the research of Bi et al.’s study, we analyzed cicRNA, miRNA and mRNA and adopted different bioinformatics analysis methods, further narrowing the range of possible molecular mechanisms [23]. In addition, we also carried out relevant experimental verification to strengthen the scientific nature of the results. In addition, different from the research of Liu et al., we chose another imRNA dataset (GSE9877, consistent with other datasets all samples were taken from the ascending aorta wall, avoiding the influence of different sampling sites on data analysis results) [24]. In addition, our experimental verification is richer, further narrowing the range of possible molecular mechanisms and strengthening the scientific nature of the results.

More and more evidence show that circRNA is widely involved in the pathogenesis and progress of diseases. Similar to long non-coding RNAs (lnc RNA), circRNA plays its role by acting as an miRNA sponge and competing endogenous RNA (ceRNA). The “ceRNA mechanism” was first proposed by Pandolfi et al. in 2011 [13]: they suggested that there is a competitive endogenous RNA (ceRNA) in the cell. These ceRNA molecules (lncRNA, circRNA, mRNA, etc.) compete for the binding sites of miRNA (miRNA response elements; MRE) to affect miRNA functioning and related mRNA expression, thereby regulating gene expression levels and affecting cell or tissue function. This competitive miRNA-binding effect is called the miRNA “sponge” effect, and the mechanism is described as an intracellular feedforward regulatory loop. Zou et al. [25] showed that hsa_circ_101238 may interact with hsa-miRNA-320a, promoting the increase of MMP-9 expression, which may lead to the aortic wall becoming fragile and thin. Marinou et al. [26] found that lncRNA H19 can be used as the ceRNA of miR-148b to enhance the expression of the target gene WNT1 in human aortic vascular smooth muscle cells (HA-VSMCs) and then trigger vascular smooth muscle cells (VSMCs) by activating the Wnt/β-catenin signaling pathway, leading to the proliferation, migration, and inhibition of apoptosis. Through the construction of CeRNA regulatory network, a total of eight circRNA-miRNA-mRNA CeRNA regulatory axes that may be closely related to aortic dissection diseases were screened in this study. Based on clinical sample research and bioinformatics analysis, we predicted that the hsa_circ_0007386/miR-1271–5P/IGF1R/AKT signal axis may play an important regulatory role in aortic dissection disease.

Our results also show that IGF1R is a key DEG associated with the pathogenesis or treatment of TAD and may be a key biomarker associated with the pathogenesis and progression of TAD. The expression levels of IGF1R were downregulated in TAD tissue from the GEO database. Similarly, the expression levels of IGF1R were downregulated in our RT-PCR data in TAD tissue. We also found that the expression level of IGF1R protein in arterial wall tissue was downregulated in TAD compared to that tissue without dissection. In addition, we found interesting that the IGF1R levels in plasma were significantly increased in TAD patients compared with healthy controls. We speculate that the downregulation of IGF1R in arterial wall tissue in TAD patients may reflexively cause the upregulated level of IGF1R in plasma. In our study, we found that PPP, which has been shown to inhibit IGF1R, can induce apoptosis in NASMCs [27]. Downregulated expression of IGF1R results in increased expression of AKT and Caspase3 in the NASMCs. Insulin-like growth factors (IGFs) are one of the important growth factors, and most of the effects of insulin-like growth factor-1 (IGF-1) are mediated through its cognate tyrosine kinase receptor, IGF1R [28]. IGF1R signaling activates multiple downstream signaling pathways, including PI3-kinase, Akt, and mitogen-activated protein kinase (MAPK). The activation of these pathways mediates different biological effects of IGF-1, including cell growth, differentiation, migration, and survival. IGF1 mRNA expression, IGF1 receptor (IGF1R), and 1GFI binding protein (IGF-1BP) are also found in the heart and blood vessels; furthermore, there may be an IGF-1–IGF1R system that has a physiological and pathophysiological regulation effect on the cardiovascular system [2,29]. Decreased levels of IGF-1 and IGF1R are associated with smooth muscle cell (SMC) apoptosis in human atherosclerotic plaques, indicating an association between downregulation of IGF-1 signaling, SMC apoptosis, and atherosclerotic burden [30]. A decrease in IGF-1 receptor density can significantly inhibit the proliferation of VSMCs [31]. Sukhanov et al. [32] showed that IGF-1 signaling in SMCs and fibroblasts is a critical determinant of normal vascular wall development and atheroprotection. Belinda et al. [33] found that moderate calcium levels increase the level of osteoprotegerin, which then increases IGF1R levels to enhance VSMC survival and block calcification. By contrast, high calcium levels lead to the inhibition of IGF1R expression and activity to further stimulate VSMC calcification. Yeap et al. found that the IGF1 system may contribute to, or be a marker for, aortic dilation in ageing men [34]. In older men, higher IGF1 and an increased ratio of IGF1/IGFBP3 are associated with AAA, while IGFBP1 is independently associated with increased aortic diameter. Mozo et al. confirmed that IGFBP-1 has been identified by a protein array approach as a potential novel biomarker of AAA [35]. Pannu et al.’s study shows that MYH11 mutations result in a distinct aortic and occlusive vascular pathology driven by IGF-1 and Ang II [36]. Estrada et al. built a novel cell signaling network model of a vascular smooth muscle cell to capture the effects of the phenotypic changes of smooth muscle cells. Their research found that the proliferation, apoptosis, and degrading activity of vascular smooth muscle cells and bistable switch are driven by positive feedback in the PI3K/AKT/mTOR signaling pathway [37]. Ying et al.’s study suggests that impaired AKT2 signaling may contribute to increased susceptibility to the development of AAD [38]. These studies further illustrate our finding that IGF1R is a key DEG associated with the pathogenesis or treatment of TAD. In addition, IGF1R/AKT signal axis involvement suggests differences in metabolism of the SMCs. Gäbel et al.’s study suggests that the metabolic response appears causatively involved in AAA progression. Clear genomic responses with activated adaptive immune responses, and particularly strong signals for metabolic switching were observed in human AAA [39]. Oller et al. found that mitochondrial dysfunction of VSMCs drives the development of aortic aneurysm in Marfan syndrome. Targeting vascular metabolism is a new available therapeutic strategy for managing aortic aneurysms associated with genetic disorders [40]. Verhagen et al. confirm that inflammatory and mitochondrial pathways play important roles in the pathophysiological processes underlying MFS-related aortic disease [41]. The mitochondrial dysfunction in these tissues and cells show a switch from oxygen-driven energy metabolism towards anaerobic energy induction from glucose into lactate. Perhaps the signature found in our study also suggests this metabolic switch.

This study had several limitations. The key hub genes and the pathways we identified have not been further confirmed by in vitro studies or other functional studies. In addition, more clinical information data were not available; thus, the significance of the expression level of key hub genes and clinical data could not be considered in this study. However, to some extent, our results could be enlightening for subsequent mechanism studies.

## 5. Conclusions

In summary, we conducted bioinformatics analysis on GEO datasets GSE97745, GSE98770, and GSE52093 from aortic wall tissues. We then constructed a ceRNA network, gene co-expression network, detected gene modules, and identified a hub gene (IGF1R), a signal axis (hsa_circ_0007386/miR-1271–5P/IGF1R) and the signaling pathways (PI3K–Akt signaling pathway) that differed between TAD and normal aorta. Finally, we identified the hsa_circ_0007386/miR-1271–5P/IGF1R/AKT signal axis, which may have crucial biological functions in the pathogenesis of TAD. In addition, IGF1R showed high diagnostic efficacy in both tissue and plasma samples in TAD, which can be considered as a diagnostic marker for TAD. We believe that our results will help researchers to further understand the possible molecular mechanism of TAD occurrence and development, and provide theoretical reference for the development of drug-assisted treatment or prevention of TAD. Further studies are needed to elucidate the potential role of the genes in this axis as diagnostic, prognostic, or therapeutic biomarkers in TAD.

## Figures and Tables

**Figure 1 biomedicines-11-00571-f001:**
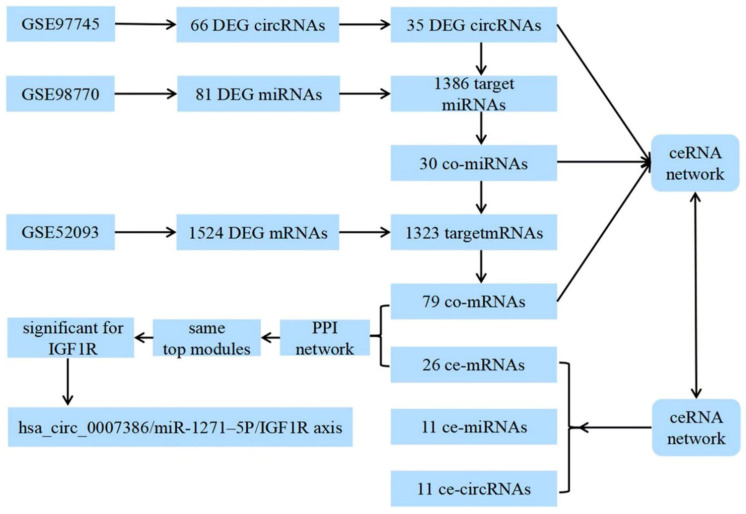
Diagram of ceRNA network construction.

**Figure 2 biomedicines-11-00571-f002:**
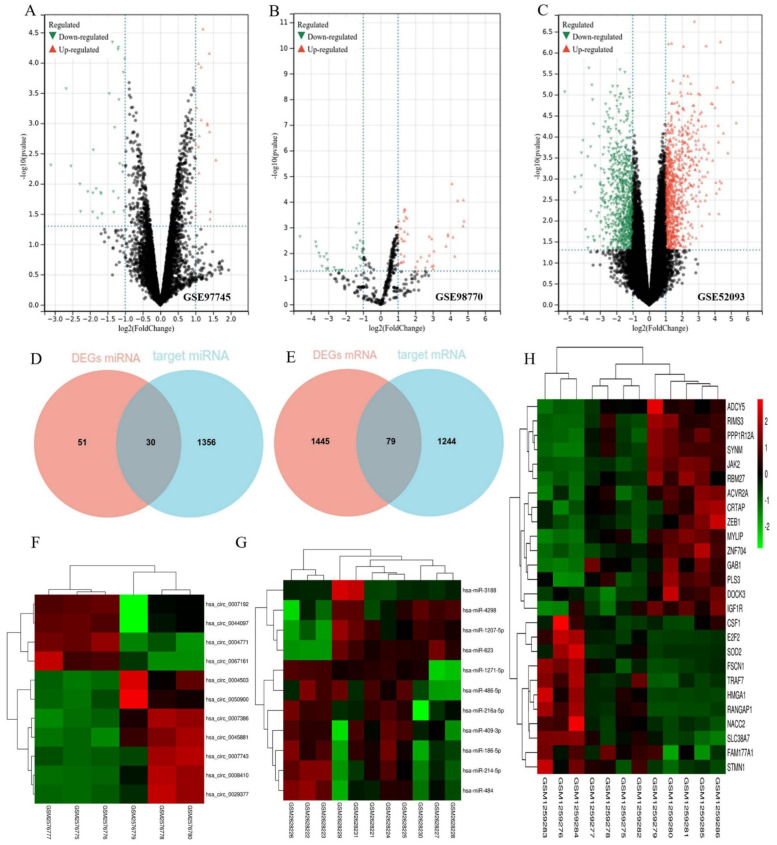
Identification of DEGs and target genes. The volcano plots show the differentially expressed circRNAs (**A**), differentially expressed miRNAs (**B**), and differentially expressed mRNAs (**C**) in the TAD and CON (normal aortic) samples. The green nodes on the left represent downregulated DEGs and the red nodes on the right represent upregulated DEGs; the black nodes represent genes with *p*-value > 0.05. Venn analysis between the differentially expressed miRNAs (**D**) and the target miRNAs (**E**). Heatmaps of 11 ce-circRNAs (**F**), 11 ce-miRNAs (**G**), and 26 ce-mRNA (**H**) from the ceRNA network used in hierarchy cluster analysis. The diagram presents the result of a two-way hierarchical clustering of all the DEGs and samples. Each row in the heatmap represents a sample, and each column represents a gene. The color scale at the right of the heatmap represents the raw Z-score ranging from green (low expression) to red (high expression).

**Figure 3 biomedicines-11-00571-f003:**
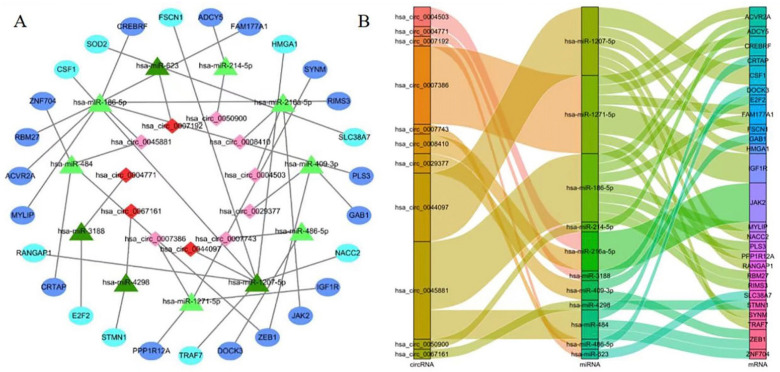
The circRNA-microRNA-mRNA network. (**A**) The ceRNA network. The oval represents mRNA, the triangle represents miRNA, and the diamond represents cicRNA. Among the corresponding shape genes, the light color represents downregulated in TAD, and the dark color represents upregulated in TAD. (**B**) Sankey diagram for the ceRNA network in TAD. Each rectangle represents a gene, and the connection degree of each gene is visualized based on the size of the rectangle.

**Figure 4 biomedicines-11-00571-f004:**
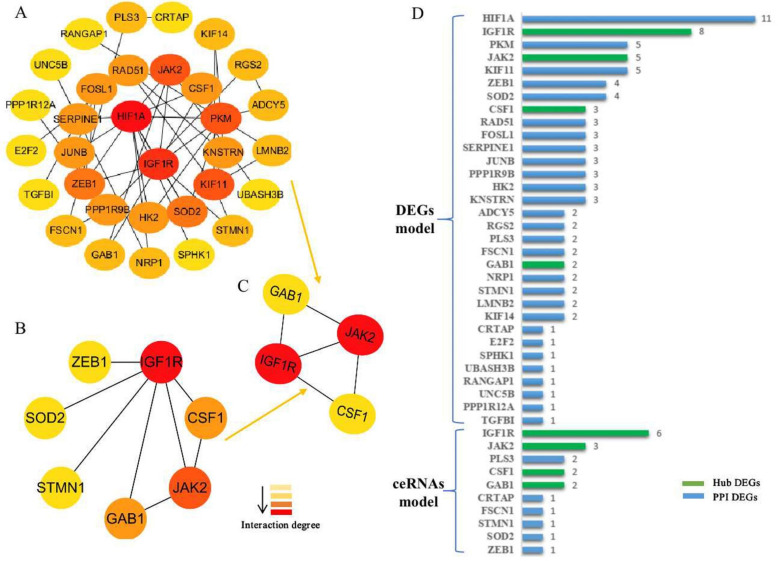
The protein–protein interaction (PPI) network constructed via the STRING database for the DEGs. (**A**) PPI network of the 79 co-mRNAs. (**B**) PPI network of the 26 ce-mRNAs. (**C**) The top modules networks were identified by Cytoscape MCODE. Both the 79 co-mRNA and the 26 ce-mRNA list showed the same top modules. The shades of color (yellow to red) represent the degree to which the molecules interact. (**D**) Genes from the DEGs (co-mRNAs and ce-mRNAs) based on the PPI network analysis with the degree of interaction in the PPI network hub nodes.

**Figure 5 biomedicines-11-00571-f005:**
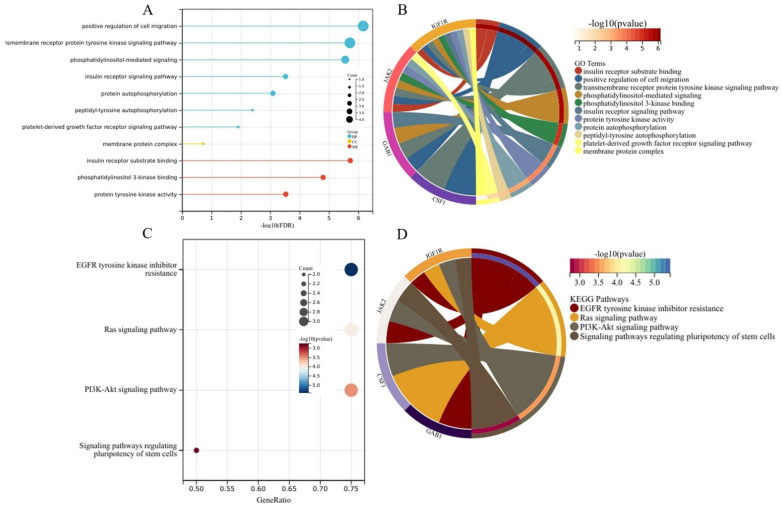
The enrichment analysis for the key module. (**A**) GO analyses of the hub nodes. GO bubble plot: the *y*-axis represents the term where genes are enriched, and the *x*-axis represents the FDR of term. The color of the dot represents the biological processes (BPs), cellular components (CCs), and molecular functions (MFs), and the size of the dot represents the number of gene enrichment. (**B**) The enriched GO terms. GO chord plot: the genes are linked via ribbons to their assigned terms. Red coding next to the selected genes indicates logFC. (**C**) KEGG bubble plot: the *y*-axis represents the pathways where genes are enriched, and the *x*-axis represents the ratio of term genes to the total genes. The color of the dot represents the logFC, and the size of the dot represents the number of gene enrichment. (**D**) The enriched KEGG pathways. KEGG chord plot: the genes are linked via ribbons to their assigned terms. Red coding next to the selected genes indicates logFC.

**Figure 6 biomedicines-11-00571-f006:**
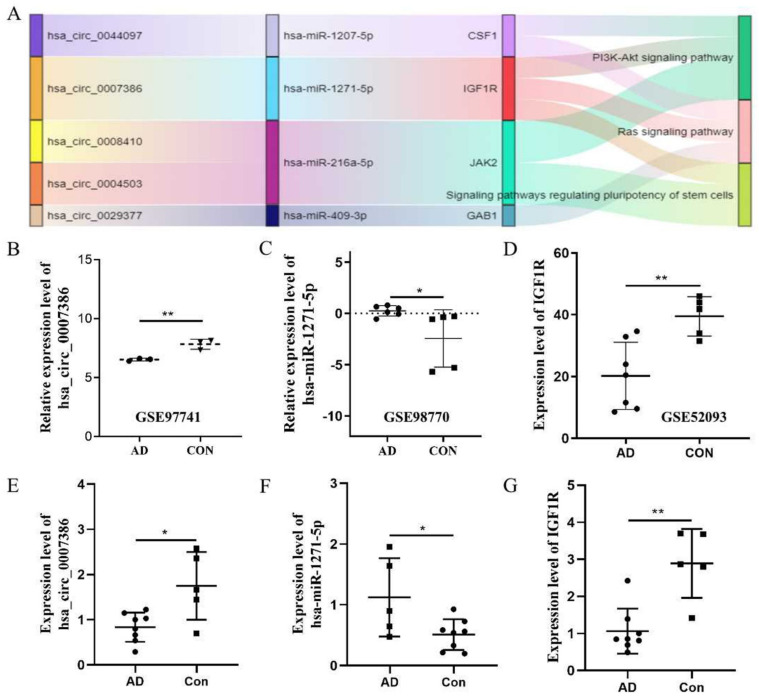
Signal axis screening and specimen verification. (**A**) Sankey diagram for the circRNA-microRNA-mRNA-pathways axis screened out in this study. Each rectangle represents a gene or pathway, and the connection degree of each gene is visualized based on the size of the rectangle. (**B**–**D**) The expression levels of the hsa-cic-0007386/miR-1271-5P/IGF1R between the TAD and CON group in the GSE97745, GSE98770, and GSE52093 datasets, respectively. * *p* < 0.05, ** *p* < 0.01. (**E**–**G**) The expression levels of the hsa-cic-0007386/miR-1271-5P/IGF1R between the TAD and CON group in our aortic dissection specimen. * *p* < 0.05, ** *p* < 0.01.

**Figure 7 biomedicines-11-00571-f007:**
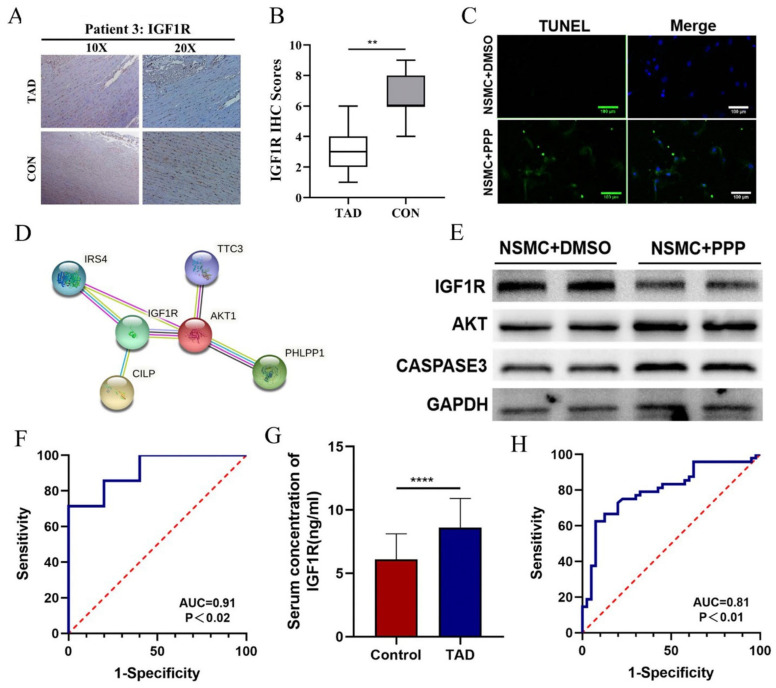
IGF1R downregulation mediates VSMC apoptosis by promoting AKT expression. (**A**) The expression of IGF1R protein in TAD aortic wall tissue and respective adjacent CON tissues were analyzed using IHC. (**B**) IGF1R expression scores are shown as box plots. (** *p* < 0.001). (**C**) Tunel stain showed PPP-induced apoptosis of NSMC. (**D**) STRING database shows IGF1R interaction with AKT. (**E**) Western blot showed that PPP-induced downregulation of IGF1R expression in NSMC further leads to the increased expression of AKT and Caspase3. (**F**) Receiver operating characteristic analysis of the sensitivity and specificity of the predictive value of the IGF1R in GSE52093. (**G**) The serum concentration of IGF1R levels between the TAD and normal patients in our data (**** *p* = 0.000). (**H**) Receiver operating characteristic analysis of the sensitivity and specificity of the predictive value of the serum concentration of IGF1R levels in our data.

**Table 1 biomedicines-11-00571-t001:** The GO enrichment analyses of differentially expressed genes between TAD and Control.

Category	Term	Description	Count	*p*-Value
BP	GO:0030335	positive regulation of cell migration	3	3.56 × 10^−4^
	GO:0048008	platelet-derived growth factor receptor signaling pathway	2	7.13 × 10^−3^
	GO:0038083	peptidyl-tyrosine autophosphorylation	2	1.39 × 10^−2^
	GO:0008286	insulin receptor signaling pathway	2	1.39 × 10^−2^
	GO:0007169	transmembrane receptor protein tyrosine kinase signaling pathway	2	1.71 × 10^−2^
	GO:0048015	phosphatidylinositol-mediated signaling	2	1.88 × 10^−2^
	GO:0046777	protein autophosphorylation	2	3.04 × 10^−2^
CC	GO:0016020	membrane	3	4.02 × 10^−2^
MF	GO:0008286	insulin receptor substrate binding	2	1.95 × 10^−3^
	GO:0043548	phosphatidylinositol 3-kinase binding	2	3.37 × 10^−3^
	GO:0004713	protein tyrosine kinase activity	2	2.35 × 10^−2^

**Table 2 biomedicines-11-00571-t002:** The Kyoto Encyclopedia of Genes and Genomes (KEGG) pathway analysis of differentially expressed genes (DEGs).

Term	Description	Count	*p*-Value	Genes
hsa04014	Ras signaling pathway	3	9.73 × 10^−5^	CSF1, GAB1, IGF1R
hsa04151	PI3K-Akt signaling pathway	3	3.43 × 10^−4^	CSF1, JAK2, IGF1R
hsa04550	Signaling pathways regulating pluripotency of stem cells	2	1.82 × 10^−3^	JAK2, IGF1R

## Data Availability

The original data of the study are available from the corresponding authors upon reasonable request.

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
