# Peer review of "Progression of Thoracic Aortic Dissection Is Aggravated by the hsa_circ_0007386/miR-1271-5P/IGF1R/AKT Axis via Induction of Arterial Smooth Muscle Cell Apoptosis"

_biomedicines, 2023, doi:10.3390/biomedicines11020571_

Round 1

Reviewer 1 Report

Progression of thoracic aortic dissection is aggravated by the hsa_circ_0007386/miR-1271-5P/IGF1R/AKT axis via induction of arterial smooth muscle cell apoptosis.

 With existing datasets, the authors analyze and combine various public datasets to mine for strongly similar RNA-driven pathways. I think this should be done more often to mine all the available data already in the public domain to enhance power of all these individual smaller datasets. However, from two of the datasets used here already a similar study has been performed. Please mention this manuscript and elaborate what is new in the current study in the intro or discussion. Bi S, Liu R, Shen Y, Gu J.J Thorac Dis. 2020 Sep;12(9):4842-4853. doi: 10.21037/jtd-20-1337.

I also have a few remarks and questions still.

 Abstract:  Our data—along with data from GSE97741, GSE98770, and GSE52093—were used to

Methods: The expression profiling datasets GSE97741 (circRNA, GPL21825), GSE98770 (mi-croRNA, GPL17660), and GSE52093 (mRNA, GPL10558) were downloaded from the GEO database (https://www.ncbi.nlm.nih.gov/geo/) of NCBI. The GSE97745 dataset…

è In both sentences there is a mistake: the GSE97741 should be GSE97745.

Has the GEO dataset GSE97745 been published in a journal? I do not see references.

Do the authors know if the patients in the GEO dataset GSE97745 and GSE52093 studies were connective tissue disease patients, such as Marfan syndrome/LDS or Ehlers Danlos syndrome or BAV patients? Or are common genetic causes for dissection excluded? The aortic signature has been described to overlap only partially between these syndromes and non-syndromic TAD patients. In the GEO GSE98770 they have excluded this specifically (Eur J Cardiothorac Surg 2017 Oct 1;52(4):810-817).

Figure 4. The color legends are not given? Red or yellow means what? Green or blue bars??

The expression levels of hsa_circ_0007386 and IGF1R were down-regulated, whereas that of miR-1271-5P was up-regulated in TAD (Figure 6B, C and D).

Similarly, the expression levels of hsa_circ_0007386 and IGF1R were down-regulated, whereas that of miR-1271-5P was up-regulated in our TAD data (Figure 6E, F and G).

This is not what I see in Figure 6C or F, is opposite. Either the conclusion is wrong or the Figure is wrong. It is also stated like this in the discussion. Please clarify.

Figure 7A,B: The detail is somewhat unfortunate, since this area contains a big tear, which cannot be calculated are area. So I am not sure how a decrease in IGFR is measured, because it is not described. Moreover, There is a lot of IGFR in the intima (I presume, so in this picture there seems to be more IGFR throughout the aortic wall. The IHC is not convincing.

Legend Fig 7E is not correct: Western blot showed PPP induced down-regulation of IGF1R expression in NSMC further leads to AKT expression increased and Caspase3 decreased.

However, I see caspase 3 increased and not decreased in the Western Blot.

The legend for Fig 7F-H is missing.

I am a bit confused in the discussion. It states: ChIP-sequencing data, but the expression data of the RNA molecules is not performed by ChIP-sequencing (chromatin immunoprecipitation-sequencing) ??

“….that PPP (a kind of specific IGF1R inhibitor) can induce..”: kind of is not scientific. Either it is specific or not. Could be rephrased by : …PPP, which has been shown to inhibit IGF1R (refs)….

Some more literature can be discussed: Eur J Endocrinol. 2012 Feb;166(2):191-7. doi: 10.1530/EJE-11-0725. And Atherosclerosis. 2012 Apr;221(2):544-50. doi: 10.1016/j.atherosclerosis.2012.01.009. and Hum Mol Genet. 2007 Oct 15;16(20):2453-62. doi: 10.1093/hmg/ddm201. And PLoS Comput Biol. 2021 Dec 13;17(12):e1009683. doi: 10.1371/journal.pcbi.1009683. and Circ Res. 2013 Feb 15;112(4):618-32. doi: 10.1161/CIRCRESAHA.112.300735.

IGF1R/Akt involvement suggests differences in metabolism of the SMCs, which has recently been found in abdominal aortic aneurysm tissue and in Marfan syndrome mice and aortic SMCs from Marfan and other aneurysm patients. Refs: J Am Heart Assoc. 2021 Sep 7;10(17):e020231. doi: 10.1161/JAHA.120.020231.  and Circulation. 2021 May 25;143(21):2091-2109. doi: 10.1161/CIRCULATIONAHA.120.051171. and Int J Mol Sci. 2021 Dec 31;23(1):438.  doi: 10.3390/ijms23010438.

The mitochondrial dysfunction in these tissues and cells show a switch from oxygen driven energy metabolism towards anaerobic energy induction from glucose into lactate. Perhaps the signature found here also suggests this metabolic switch, which can be discussed as well to provide more in depth understanding of the data.

Author Response

Dear editors and reviewers:

We are grateful to you for your valuable comments and suggestions, which have helped us a lot to improve the quality of our manuscript. We have studied the comments carefully and made corrections according to your helpful advice and detailed suggestions. We hope all these revisions would meet your requirements. Below are the details of our revisions.

Comments and Suggestions for Authors

Progression of thoracic aortic dissection is aggravated by the hsa_circ_0007386/miR-1271-5P/IGF1R/AKT axis via induction of arterial smooth muscle cell apoptosis.

With existing datasets, the authors analyze and combine various public datasets to mine for strongly similar RNA-driven pathways. I think this should be done more often to mine all the available data already in the public domain to enhance power of all these individual smaller datasets. However, from two of the datasets used here already a similar study has been performed. Please mention this manuscript and elaborate what is new in the current study in the intro or discussion. Bi S, Liu R, Shen Y, Gu J.J Thorac Dis. 2020 Sep;12(9):4842-4853. doi: 10.21037/jtd-20-1337.

Response: Thank you for your valuable comments and suggestions. Bi et al. found some possible interactive networks of miRNAs and genes which tend to play an important role in the pathogenesis of AAD. Different from the research of Bi et al., we analyzed cicRNA, miRNA and mRNA and adopted different bioinformatics analysis methods, further narrowing the range of possible molecular mechanisms. In addition, we also carried out relevant experimental verification to strengthen the scientific nature of the results. We have added this part into the manuscript.

Modifications to Page 10, Line 10-13: Different from the research of Bi et al., we analyzed cicRNA, miRNA and mRNA and adopted different bioinformatics analysis methods, further narrowing the range of possible molecular mechanisms. In addition, we also carried out relevant experimental verification to strengthen the scientific nature of the results.

I also have a few remarks and questions still.

Abstract:  Our data—along with data from GSE97741, GSE98770, and GSE52093—were used to

Methods: The expression profiling datasets GSE97741 (circRNA, GPL21825), GSE98770 (mi-croRNA, GPL17660), and GSE52093 (mRNA, GPL10558) were downloaded from the GEO database (https://www.ncbi.nlm.nih.gov/geo/) of NCBI. The GSE97745 dataset…è In both sentences there is a mistake: the GSE97741 should be GSE97745.

Response: Thank you for your valuable comments and suggestions. I apologize for our spelling mistakes. We have corrected them in the manuscript and have carefully checked the spelling of all manuscripts again.

Has the GEO dataset GSE97745 been published in a journal? I do not see references.

Response: Thank you for your valuable comments. We found an article related to GSE97745, (Liu De-Bin,He You-Fu,Chen Gui-Jian et al.Int J Gen Med, 2022, 15: 3951-3964). What is different in our study is that we chose another imRNA dataset (GSE9877, consistent with other datasets all samples were taken from the ascending aorta wall, avoiding the influence of different sampling sites on data analysis results). In addition, our experimental verification is richer, further narrowing the range of possible molecular mechanisms and strengthening the scientific nature of the results. We have added this part to the manuscript.

Modifications to Page 10, Line 14-18: Different from the research of Liu et al., we chose another imRNA dataset (GSE9877, consistent with other datasets all samples were taken from the ascending aorta wall, avoiding the influence of different sampling sites on data analysis results). In addition, our experimental verification is richer, further narrowing the range of possible molecular mechanisms and strengthening the scientific nature of the results.

Do the authors know if the patients in the GEO dataset GSE97745 and GSE52093 studies were connective tissue disease patients, such as Marfan syndrome/LDS or Ehlers Danlos syndrome or BAV patients? Or are common genetic causes for dissection excluded? The aortic signature has been described to overlap only partially between these syndromes and non-syndromic TAD patients. In the GEO GSE98770 they have excluded this specifically (Eur J Cardiothorac Surg 2017 Oct 1;52(4):810-817).

Response: Thank you for your valuable comments and suggestions. We contacted the data uploader through email and know that the GSE97745 and GSE52093 have excluded these patients (such as Marfan syndrome/LDS or Ehlers Danlos syndrome or BAV patients).

Figure 4. The color legends are not given? Red or yellow means what? Green or blue bars??

Response: Thank you for your valuable comments and suggestions. We have added color legends in Figure 4.

Modifications in Figure 4:Please see the attachment.

The expression levels of hsa_circ_0007386 and IGF1R were down-regulated, whereas that of miR-1271-5P was up-regulated in TAD (Figure 6B, C and D).

Similarly, the expression levels of hsa_circ_0007386 and IGF1R were down-regulated, whereas that of miR-1271-5P was up-regulated in our TAD data (Figure 6E, F and G).

This is not what I see in Figure 6C or F, is opposite. Either the conclusion is wrong or the Figure is wrong. It is also stated like this in the discussion. Please clarify.

Response: Thank you for your valuable comments and suggestions. I apologize for our mistake. We marked the abscissa of Figure 6F upside down, and we have corrected it.

Modifications in Figure 6F:Please see the attachment.

Figure 7A,B: The detail is somewhat unfortunate, since this area contains a big tear, which cannot be calculated are area. So I am not sure how a decrease in IGFR is measured, because it is not described. Moreover, There is a lot of IGFR in the intima (I presume, so in this picture there seems to be more IGFR throughout the aortic wall. The IHC is not convincing.

Response: Thank you for your valuable comments and suggestions. I apologize fo the image we chose are not typical enough. In the TAD patient in Figure 7A, the intima was torn, part of the blood entered the intima, resulting in increased IGF1R level in the intima, which was also consistent with our other findings. We found an interesting phenomenon when we performed ELISA tests on plasma from TAD patients. The IGF1R levels in plasma were significantly increased in TAD patients compared with healthy controls. We speculate that the down-regulation of IGF1R in arterial wall tissue in TAD patients may reflexively cause the up-regulated level of IGF1R in plasma. Of course, further experimental data is needed to confirm our conclusions. Therefore, in order to avoid the influence of intimal tearing on the experimental results, only the expression level of IGF1R in the medial membrane was calculated when detecting the level of IGF1R in the arterial wall tissue. We have supplemented the immunohistochemical scoring method of this study in the manuscript, also replaced the more typical immunohistochemical image in the Figure 7A.

Modifications in Figure 7A:Please see the attachment.

Legend Fig 7E is not correct: Western blot showed PPP induced down-regulation of IGF1R expression in NSMC further leads to AKT expression increased and Caspase3 decreased.

However, I see caspase 3 increased and not decreased in the Western Blot.

Response: Thank you for your valuable comments and suggestions. I apologize for our spelling error, which has been corrected in Legend Fig 7E and the manuscript spelling has been carefully checked again.

Modifications in Legend Fig 7E: Western blot showed PPP induced down-regulation of IGF1R expression in NSMC further leads to the expression of AKT and Caspase3 increased.

The legend for Fig 7F-H is missing.

Response: Thank you for your valuable comments and suggestions. We have added legends in Fig 7F-H.

Modifications in Legend Fig 7F-H: (F) Receiver operating characteristic analysis of the sensitivity and specificity of the predictive value of the IGF1R in GSE52093. (G) The serum concentration of IGF1R levels between the TAD and normal patients in our data. (H) Receiver operating characteristic analysis of the sensitivity and specificity of the predictive value of the serum concentration of IGF1R leve in our data.

I am a bit confused in the discussion. It states: ChIP-sequencing data, but the expression data of the RNA molecules is not performed by ChIP-sequencing (chromatin immunoprecipitation-sequencing) ??

Response: Thank you for your valuable comments and suggestions. All analyzed in this paper are high-throughput sequencing data on GEO database. I apologize for our spelling mistakes. We have corrected them in the manuscript and have carefully checked the spelling of all manuscripts again.

“….that PPP (a kind of specific IGF1R inhibitor) can induce..”: kind of is not scientific. Either it is specific or not. Could be rephrased by : …PPP, which has been shown to inhibit IGF1R (refs)….

Response: Thank you for your valuable comments and suggestions. We have corrected “In our study, we found that PPP (a kind of specific IGF1R inhibitor) can induce apoptosis in NASMCs.” to “In our study, we found that PPP, which has been shown to inhibit IGF1R, can induce apoptosis in NASMCs.” in the manuscript.

Modifications to Page 11, Line 21-22: In our study, we found that PPP, which has been shown to inhibit IGF1R, can induce apoptosis in NASMCs.

Some more literature can be discussed: Eur J Endocrinol. 2012 Feb;166(2):191-7. doi: 10.1530/EJE-11-0725. And Atherosclerosis. 2012 Apr;221(2):544-50. doi: 10.1016/j.atherosclerosis.2012.01.009. and Hum Mol Genet. 2007 Oct 15;16(20):2453-62. doi: 10.1093/hmg/ddm201. And PLoS Comput Biol. 2021 Dec 13;17(12):e1009683. doi: 10.1371/journal.pcbi.1009683. and Circ Res. 2013 Feb 15;112(4):618-32. doi: 10.1161/CIRCRESAHA.112.300735.

Response: Thank you for your valuable comments and suggestions. Thank you for your helpful literature, which is very helpful to increase the scientific and rigorous of our research. We have added this part to the manuscript.

Modifications to Page 12, Line 14-24: Yeap et al found that the IGF1 system may contribute to, or be a marker for, aortic dilation in ageing men. In older men, higher IGF1 and an increased ratio of IGF1/IGFBP3 are associated with AAA, while IGFBP1 is independently associated with increased aortic diameter. Mozo et al confirmed that IGFBP-1 has been identified by a protein array approach as a potential novel biomarker of AAA. Pannu et al’ study shows that MYH11 mutations result in result in a distinct aortic and occlusive vascular pathology driven by IGF-1 and Ang II. Estrada et al built a novel cell signaling network model of a vascular smooth muscle cell to capture the effects of the phenotypic changes of smooth muscle cells. Their research found that the proliferation, apoptosis, and degrading activity of vascular smooth muscle cells and bistable switch drive by positive feedback in the PI3K/AKT/mTOR signaling pathway. Ying et al’ study suggest that impaired AKT2 signaling may contribute to increased susceptibility to the development of AAD.

IGF1R/Akt involvement suggests differences in metabolism of the SMCs, which has recently been found in abdominal aortic aneurysm tissue and in Marfan syndrome mice and aortic SMCs from Marfan and other aneurysm patients. Refs: J Am Heart Assoc. 2021 Sep 7;10(17):e020231. doi: 10.1161/JAHA.120.020231.  and Circulation. 2021 May 25;143(21):2091-2109. doi: 10.1161/CIRCULATIONAHA.120.051171. and Int J Mol Sci. 2021 Dec 31;23(1):438.  doi: 10.3390/ijms23010438.

The mitochondrial dysfunction in these tissues and cells show a switch from oxygen driven energy metabolism towards anaerobic energy induction from glucose into lactate. Perhaps the signature found here also suggests this metabolic switch, which can be discussed as well to provide more in depth understanding of the data.

Response: Thank you for your valuable suggestions, which is very helpful to increase the scientific and rigorous of our research. We added the following into the discussion section:

Modifications to Page 13, Line 2-13: In addition, IGF1R/AKT signal axis involvement suggests differences in metabolism of the SMCs. Gäbel et al’ study suggest that the metabolic response appears causatively involved in AAA progression. Clear genomic responses with activated adaptive immune responses, and particularly strong signals for metabolic switching were observed in human AAA. Oller et al found that mitochondrial dysfunction of VSMCs drives the development of aortic aneurysm in Marfan syndrome. Targeting vascular metabolism is a new available therapeutic strategy for managing aortic aneurysms associated with genetic disorders. Verhagen et al confirm that inflammatory and mitochondrial pathways play important roles in the pathophysiological processes underlying MFS-related aortic disease.The mitochondrial dysfunction in these tissues and cells show a switch from oxygen driven energy metabolism towards anaerobic energy induction from glucose into lactate. Perhaps the signature found in our study also suggests this metabolic switch.

In conclusion, we have checked the manuscript and revised it according to all the comments. We submit here the revised manuscript as well as a list of changes. If you have any question about this manuscript, please do not hesitate to let me know.

Sincerely yours.

Reviewer 2 Report

I enjoyed reading the paper and the data was well presented

I have a number of questions.

How were the GSE97745, GSE98770 and GSE52093 chosen? and how were the 35 DEG circRNAs selected for further study?

IGF1R and AKT pathways have previously been studied in various forms of atherosclerosis, Instent stenosis, metabolic syndrome etc...It is unfortunate that the clinical data on the patients studied was not available. If even the clinical data on the small subset of 12 patients on whom RT-PCR of the aortic tissue and serum was obtained it may have given us some insight into how various risk factors including age, hypertension, family history, etc may be relevant.

At what time point was the analysis performed on the TAD tissue and serum? I assume it was at the time of emergency surgery for the dissection? How does this influence gene expression, and does it influence the impact of IGF1R as a biomarker to predict dissection, i.e. as opposed to being present after dissection? How much SMC apoptosis occurred pre-injury and as result of injury to the media from the dissection? 

Author Response

Dear editors and reviewers:

We are grateful to you for your valuable comments and suggestions, which have helped us a lot to improve the quality of our manuscript. We have studied the comments carefully and made corrections according to your helpful advice and detailed suggestions. We hope all these revisions would meet your requirements. Below are the details of our revisions.

Comments and Suggestions for Authors

I enjoyed reading the paper and the data was well presented

I have a number of questions.

How were the GSE97745, GSE98770 and GSE52093 chosen? and how were the 35 DEG circRNAs selected for further study?

Response: Thank you for your valuable comments and suggestions. The sequencing data of GSE97745, GSE98770 and GSE5209 were all derived from arterial tissue samples. All samples were taken from the ascending aorta wall, avoiding the influence of different sampling sites on data analysis results.We used CSCD (http://gb.whu.edu.cn/CSCD/) and circBase (http://www.circbase.org/) to predict miRNA-binding sites (MREs).Here we only retain 35 DEG circRNAs with miRNA-binding sites.

IGF1R and AKT pathways have previously been studied in various forms of atherosclerosis, Instent stenosis, metabolic syndrome etc...It is unfortunate that the clinical data on the patients studied was not available. If even the clinical data on the small subset of 12 patients on whom RT-PCR of the aortic tissue and serum was obtained it may have given us some insight into how various risk factors including age, hypertension, family history, etc may be relevant.

At what time point was the analysis performed on the TAD tissue and serum? I assume it was at the time of emergency surgery for the dissection? How does this influence gene expression, and does it influence the impact of IGF1R as a biomarker to predict dissection, i.e. as opposed to being present after dissection?

Response:Thank you for your valuable comments and precise suggestions. As the reviewer said, the analysis performed on the TAD tissue and serum at the time of emergency surgery for the dissection. Based on the current status of the diagnosis and treatment of TAD, many patients are found when they come to see a doctor after having symptoms related to TAD (such as chest and back pain). In many research centers, the existing tissue samples are also collected during emergency surgery, which is also the limitation of scientific research at present. The influence the impact of IGF1R as a biomarker to predict dissection, to a certain extent, can be further elucidated by animal experiments, which is one of the tasks of our further research in the future. In addition, we will delve further and try to screen for these biomarkers in asymptomatic individuals by the peripheral blood circulating factors. After all, fom a practical standpoint it would be much easier to get a blood sample than a tissue sample.

How much SMC apoptosis occurred pre-injury and as result of injury to the media from the dissection?

Response: Thank you for your valuable comments and precise suggestions. The injury to the media from the dissection also caused a certain degree of SMC apoptosis, can be further elucidated by animal experiments, which is one of the tasks of our further research in the future.

In conclusion, we have checked the manuscript and revised it according to all the comments. We submit here the revised manuscript as well as a list of changes. If you have any question about this manuscript, please do not hesitate to let me know.

Sincerely yours.

Reviewer 3 Report

1.     Introduction, page 2: The mortality data refer to type A dissections. The authors should distinguish between type A and B aortic dissections as their prognoses are different.

2.     Materials and methods, page 2: The authors state that samples were taken from normal and TAD aortas but do not describe how these samples were obtained. Please clarify.

3.     Discussion: Do the biomarkers that were evaluated require a tissue sample or are they circulating factors as well. From a practical standpoint it would be much harder to get a tissue sample than a blood sample.

4.     Discussion: What are the therapeutic implications of these findings. Can any genetic therapy be done to reduce the risk of TAD? Is screening for these biomarkers in asymptomatic individuals feasible.

Author Response

Dear editors and reviewers:

We are grateful to you for your valuable comments and suggestions, which have helped us a lot to improve the quality of our manuscript. We have studied the comments carefully and made corrections according to your helpful advice and detailed suggestions. We hope all these revisions would meet your requirements. Below are the details of our revisions.

Comments and Suggestions for Authors

  1. Introduction, page 2: The mortality data refer to type A dissections. The authors should distinguish between type A and B aortic dissections as their prognoses are different.

Response: Thank you for your valuable comments and suggestions. We have revised this part and added it to the manuscript after re-quoting relevant literature.

Modifications to Page 3, Line 10-14: If AD occurs within the type A dissections, 40% of patients die immediately and mortality is 1%–2% for each hour afterwards resulting in a 48-hour mortality of approximately 50%. In type B dissections, approximately 36%–50% of patients with thoracic aortic dissection (TAD) will die of aortic rupture within 48 hours, and 65%–75% will die within 2 weeks; the 1-year survival rate is less than 10%.

  1. Materials and methods,page 2: The authors state that samples were taken from normal and TAD aortas but do not describe how these samples were obtained. Please clarify.

Response: Thank you for your valuable comments and suggestions. We have added this to the methods section.

Modifications to Page 5, Line 13-17: The experimental samples in our center were taken from aortic walls and blood samples of TAD patients and organ donors. The analysis performed on the TAD tissue and serum at the time of emergency surgery for the dissection. Aortic specimens were collected from type A TAD patients undergoing aortic replacement in the intimal tear position at the hospital. Normal thoracic aortas (NA) were obtained from organ donors without aortic diseases.

  1. Discussion: Do the biomarkers that were evaluated require a tissue sample or are they circulating factors as well. From a practical standpoint it would be much harder to get a tissue sample than a blood sample.

Response: Thank you for your valuable comments and suggestions. The samples used in the GEO database we obtained are tissue samples. The data we used for experimental verification were tissue sample and blood sample, and the results of our tissue sample and blood sample had good consistency.

  1. Discussion: What are the therapeutic implications of these findings. Can any genetic therapy be done to reduce the risk of TAD? Is screening for these biomarkers in asymptomatic individuals feasible.

Response: Thank you for your valuable comments and suggestions. Our results help researchers to further understand the possible molecular mechanism of TAD occurrence and development, and provide theoretical reference for the development of drug-assisted treatment or prevention of TAD. In addition, we will delve further and try to screen for these biomarkers in asymptomatic individuals by the peripheral blood circulating factors. After all, fom a practical standpoint it would be much easier to get a blood sample than a tissue sample. We have integrated and supplemented this part content into the manuscript.

Modifications to Page 14, Line 3-5:We believe that our results help researchers to further understand the possible molecular mechanism of TAD occurrence and development, and provide theoretical reference for the development of drug-assisted treatment or prevention of TAD.

In conclusion, we have checked the manuscript and revised it according to all the comments. We submit here the revised manuscript as well as a list of changes. If you have any question about this manuscript, please do not hesitate to let me know.

Sincerely yours.

Reviewer 4 Report

The paper written by the following Authors: Xinsheng Xie, Xiang Hong, Shichai Hong, Yulong Huang, Gang Chen, Yihui Chen, Yue Lin, Weifeng Lu, Weiguo Fu, and Lixin Wang entitled “Progression of thoracic aortic dissection is aggravated by the hsa_circ_0007386/miR-1271-5P/IGF1R/AKT axis via induction of arterial smooth muscle cell apoptosis” presents an interesting study on o identification and verification the key ceRNA networks with clinical value for the treatment of thoracic aortic dissection.

Although the paper is interesting, I have some major concerns:

Title

The title reflects the results presented here.

Material and Methods

1. There is no information about applied equipment for the experiments. It should be included in the manuscript.

2. This part “Immunohistochemistry (IHC), Immunofluorescence and western blot assay” should be explained in more details. It is not enough to indicate only two references.

Author Response

Dear editors and reviewers:

We are grateful to you for your valuable comments and suggestions, which have helped us a lot to improve the quality of our manuscript. We have studied the comments carefully and made corrections according to your helpful advice and detailed suggestions. We hope all these revisions would meet your requirements. Below are the details of our revisions.

Comments and Suggestions for Authors

The paper written by the following Authors: Xinsheng Xie, Xiang Hong, Shichai Hong, Yulong Huang, Gang Chen, Yihui Chen, Yue Lin, Weifeng Lu, Weiguo Fu, and Lixin Wang entitled “Progression of thoracic aortic dissection is aggravated by the hsa_circ_0007386/miR-1271-5P/IGF1R/AKT axis via induction of arterial smooth muscle cell apoptosis” presents an interesting study on o identification and verification the key ceRNA networks with clinical value for the treatment of thoracic aortic dissection.

Although the paper is interesting, I have some major concerns:

Title

The title reflects the results presented here.

Material and Methods

  1. There is no information about applied equipment for the experiments. It should be included in the manuscript.

Response: Thank you for your valuable comments and suggestions. We have supplemented this part in the manuscript.

Modifications to Page 5, Line 1-5:

Reagents and equipment

Antibodies against IGF1R, AKT and CASPASE3 were purchased from Abcam (https://www.abcam.cn/). Anti-GAPDH antibodies were from Cell Signaling Technology. Secondary antibodies ( goat anti-rabbit Alexa 488 and 594) used for immunocytochemistry were from Life Technologies. ELISA kit and TUNEL Assay Kits from Abcam.Total RNA was extracted from tissue samples using TRIzol® reagent (Invitrogen; Thermo Fisher Scientific, Inc.).

  1. This part “Immunohistochemistry (IHC), Immunofluorescence and western blot assay” should be explained in more details. It is not enough to indicate only two references.

Response: Thank you for your valuable comments and suggestions. We have supplemented this part of experimental methods in detail in the manuscript.

Modifications to Page 6, Line 21-25: Immunohistochemistry (IHC) assays and evaluation of immunohistochemistry intensity were performed as previously described. TUNEL staining assay was performed according to the method steps provided by the TUNEL Assay Kits (ab66108, abcam, USA). Western blot assays were performed as previously described. Detailed methods are available in the supplementary files.

Immunohistochemistry (IHC)

Paraffin sections containing sufficient formalin fixed arterial wall tissue were sectioned continuously at a thickness of 4μm and were mounted on silage coated slides for immunohistochemical analysis. The slices were deparaffinized with xylene and rehydrated in 95%, 85% and 75% ethanol. Antigen retrieval was performed by subjecting the slides to high-pressure sterilization at 121°C for 2 min in 0.01 mol/L sodium citrate buffer solution (pH 6.0). Endogenous peroxidase activity was blocked by incubating the slides with 3% H2O2 at room temperature for 10 min. The slices were then washed in phosphate buffered saline (PBS) solution and blocked in 10% goat serum (Zhongshan Biotechnology Co, Ltd.) for 30 minutes. Next, the sections were incubated with diluted rabbit anti-human IGF1R (ab263903 diluted 1: 500.USA) overnight in a humidified chamber at 4°C. After three washes in PBS, the sections were incubated with the secondary antibody conjugated to horseradish peroxidase at room temperature for 30 minutes. The signal was developed with diaminobenzidine solution, which was followed by counterstaining in 20% hematoxylin. Finally, all slides were dehydrated and mounted on cover glass. For negative controls, antibody diluent was substituted for the primary antibody.

Western blot

Total protein was extracted from arterial wall tissue with RIPA lysis buffer (Biyotime, China) containing protease inhibitors. The protein concentration of the lysates was analyzed by BCA protein assays (Thermo Fisher Scientific, USA). 40 μg of protein was separated on a 10% SDS-polyacrylamide gel and blotted onto polyvinylidene difluoride (PVDF) membranes (Millipore, USA). After blocking with 5% bovine serum albumin (BSA) for 1 h, the membranes were then incubated with primary antibodies [IGF1R (ab263903, Abcam, USA); AKT (ab8805, abcam, USA); CASPASE3 (ab4051, Abcam, USA)] overnight at 4°C and horseradish peroxidase-conjugated secondary antibodies for 1 h at room temperature. Immunoreactive signals were detected using the ECL detection system. Immunoblotting of glyceraldehyde-3-phosphate dehydrogenase (GAPDH) was performed as an internal control.

In conclusion, we have checked the manuscript and revised it according to all the comments. We submit here the revised manuscript as well as a list of changes. If you have any question about this manuscript, please do not hesitate to let me know.

Sincerely yours.

Reviewer 5 Report

In this interesting article, the authors tried to investigate the molecular mechanisms associated with thoracic aortic dissection (TAD). In details, the purpose of this study was to identify and verify the key ceRNA networks with clinical value for the treatment of TAD. They were able to identify the hsa_circ_0007386/miR-1271–5P/IGF1R/AKT signal axis, which may have crucial biological functions in the pathogenesis of TAD, and in addition, IGF1R showed high diagnostic efficacy in both tissue and plasma samples in TAD, which can be considered as a diagnostic marker for TAD.

The study seems to be well conducted and the results are clearly presented.

From the point of view of a cardiac surgeon as I am, this paper is really interesting but completely lacks of clinical relevance. Consequently, the purpose of the study cited by the authors “identify and verify the key ceRNA networks with clinical value” has not been reached. On the other hand, the authors cite the lack of clinical information between the limitations of the study. My suggestion would be to change the declared purpose of the study and not try to find any clinical relevance from this work. 

Moreover, I think the authors should be encouraged to continue their work and I expect them to explore in further studies the clinical relevance of what they discovered through bioinformatics analysis that could improve our management of a pathology that is still a challenge for surgeons

Author Response

Dear editors and reviewers:

We are grateful to you for your valuable comments and suggestions, which have helped us a lot to improve the quality of our manuscript. We have studied the comments carefully and made corrections according to your helpful advice and detailed suggestions. We hope all these revisions would meet your requirements. Below are the details of our revisions.

Comments and Suggestions for Authors

In this interesting article, the authors tried to investigate the molecular mechanisms associated with thoracic aortic dissection (TAD). In details, the purpose of this study was to identify and verify the key ceRNA networks with clinical value for the treatment of TAD. They were able to identify the hsa_circ_0007386/miR-1271–5P/IGF1R/AKT signal axis, which may have crucial biological functions in the pathogenesis of TAD, and in addition, IGF1R showed high diagnostic efficacy in both tissue and plasma samples in TAD, which can be considered as a diagnostic marker for TAD.

The study seems to be well conducted and the results are clearly presented.

From the point of view of a cardiac surgeon as I am, this paper is really interesting but completely lacks of clinical relevance. Consequently, the purpose of the study cited by the authors “identify and verify the key ceRNA networks with clinical value” has not been reached. On the other hand, the authors cite the lack of clinical information between the limitations of the study. My suggestion would be to change the declared purpose of the study and not try to find any clinical relevance from this work.

Moreover, I think the authors should be encouraged to continue their work and I expect them to explore in further studies the clinical relevance of what they discovered through bioinformatics analysis that could improve our management of a pathology that is still a challenge for surgeons

Response: Thank you for your valuable comments and suggestions. This study has not been able to obtain enough clinical data to prove the clinical relevance of our research conclusions, which is also the shortcoming of our study. In addition, we also haved revised the purpose of the study to “ The purpose of this study was to identify and verify the key ceRNA networks which may have crucial biological functions in the pathogenesis of TAD.” In our future work, we will further collect more clinical data to confirm and discuss the clinical value of our findings in the diagnosis and treatment of TAD.

Modifications to Page 2, Line 5-6: The purpose of this study was to identify and verify the key ceRNA networks which may have crucial biological functions in the pathogenesis of TAD.

In conclusion, we have checked the manuscript and revised it according to all the comments. We submit here the revised manuscript as well as a list of changes. If you have any question about this manuscript, please do not hesitate to let me know.

Sincerely yours.

Round 2

Reviewer 4 Report

I accept the manuscriptin the present form.